**Data Availability Statement:** All relevant data are within the manuscript. All study data remain the

# Performance of a novel rapid test for recent HIV infection among newly-diagnosed pregnant adolescent girls and young women in four high-HIV-prevalence districts—Malawi, 2017–2018

Elfriede A. Agyemang[1]*, Andrea A. Kim[1], Trudy Dobbs[1], Innocent Zungu[2], Danielle Payne[2], Andrew D. Maher[3], Kathryn Curran[1], Evelyn Kim[2], Hastings Kwalira[4], Henry Limula[4], Amitabh Adhikari[1], Susie Welty[3], James Kandulu[4], Rose Nyirenda[4], Andrew F. Auld[2], George W. Rutherford[3], Bharat S. Parekh[1]*

1 Division of Global HIV and TB, US Centers for Disease Control and Prevention, Atlanta, GA, United States of America, 2 Division of Global HIV and TB, US Centers for Disease Control and Prevention, Lilongwe, Malawi, 3 Institute for Global Health Sciences, University of California, San Francisco, San Francisco, CA, United States of America, 4 Ministry of Health and Population, Lilongwe, Malawi

* eagyemang@cdc.gov (EAA); bsp1@cdc.gov (BSP)

## Abstract

Tests for recent HIV infection (TRI) distinguish recent from long-term HIV infections using markers of antibody maturation. The limiting antigen avidity enzyme immunoassay (LAg EIA) is widely used with HIV viral load (VL) in a recent infection testing algorithm (RITA) to improve classification of recent infection status, estimate population-level HIV incidence, and monitor trends in HIV transmission. A novel rapid test for recent HIV infection (RTRI), Asanté™, can determine HIV serostatus and HIV recency within minutes on a lateral flow device through visual assessment of test strip or reader device. We conducted a field-based laboratory evaluation of the RTRI among pregnant adolescent girls and young women (AGYW) attending antenatal clinics (ANC) in Malawi. We enrolled pregnant AGYW aged <25 years testing HIV-positive for the first time at their first ANC visit from 121 ANCs in four high-HIV burden districts. Consenting participants provided blood for recency testing using LAg EIA and RTRI, which were tested in central laboratories. Specimens with LAg EIA normalized optical density values ≤2.0 were classified as probable recent infections. RTRI results were based on: (1) visual assessment: presence of a long-term line (LT) indicating non-recent infection and absence of the line indicating recent infection; or (2) a reader; specimens with LT line intensity units <3.0 were classified as probable recent infections. VL was measured for specimens classified as a probable recent infections by either assay; those with HIV-1 RNA ≥1,000 copies/mL were classified as confirmed recent infections. We evaluated the performance of the RTRI by calculating correlation between RTRI and LAg EIA results, and percent agreement and kappa between RTRI and LAg EIA RITA results. Between November 2017 to June 2018, 380 specimens were available for RTRI evaluation; 376 (98.9%) were confirmed HIV-positive on RTRI. Spearman's rho between RTRI and LAg

property of Malawi Ministry of Health and can be requested by contacting the HTS Team at the Malawi Department of HIV and AIDS (mailto: hts@hivmw.org).

**Funding:** This study was supported by the President's Emergency Plan for AIDS Relief (PEPFAR) through the Centers for Disease Control and Prevention (CDC) under the terms of cooperative agreement U2GGH000977. The findings and conclusions in this report are those of the authors and do not necessarily represent the official position of the funding agencies.

**Competing interests:** The authors have read the journal's policy and have the following competing interests to declare: As an inventor of LAg EIA and rapid test for recent infection, BSP receives a portion of royalties from the sale of these tests as per policy of the United States government. There are no patents, products in development or marketed products associated with this research to declare. However, the product (Asante Rapid Recency Assay kit) was developed and commercialized under a licensing agreement with the CDC. This does not alter our adherence to PLOS ONE policies on sharing data and materials.

EIA was 0.72 indicating strong correlation. Percent agreement and kappa between RTRI- and LAg EIA-based RITAs were >90% and >0.65 respectively indicating substantial agreement between the RITAs.This was the first field evaluation of an RTRI in sub-Saharan Africa, which demonstrated good performance of the assay and feasibility of integrating RTRI into routine HIV testing services for real-time surveillance of recent HIV infection.

## Introduction

In an effort to accelerate the global AIDS response to end the HIV epidemic, the Joint United Nations Programme on HIV/AIDS (UNAIDS) set an ambitious target in 2014 for 90% of people living with HIV (PLHIV) to know their HIV status, 90% of those PLHIV diagnosed to be on treatment, and 90% of those on treatment to have viral suppression ("90-90-90" target) by 2020 [1]. In addition, a fast-track goal of reducing the levels of new HIV infections among adults in 2010 by 75% to 500,000 annually by 2020 was also set [2]. With this mandate, countries demonstrated a strong will and commitment to end the epidemic by rapidly adopting and scaling up new strategies in their HIV policies and programming [3–5]. In 2018, an estimated 37.9 million people in the world were living with HIV, of whom 23 million people were on antiretroviral therapy (ART), an increase from 7.7 million in 2010; and 770,000 people died from AIDS-related illnesses, a decline by 33% since 2010. New HIV infections had been reduced by 40% since their peak in 1997 and by 16% since 2010 to 1.7 million in 2018 [6,7]. Despite these significant advances toward the 90-90-90 targets, there remain gaps that call for innovative approaches in order to end the epidemic. For example, the reduction in the annual number of new HIV infections is occurring at a slower pace than what is needed to reach the goal of less than 50,000 new infections per year by 2020 [8], and there is a possibility of epidemic rebound with the current rate of population growth without novel global HIV response strategies [9]. Monitoring and understanding the epidemiology and distribution of new and recent HIV infections can inform effective HIV prevention programs [10].

Antibody-based tests for recent infection (TRI), which are based on the ability to measure biomarkers detected in the early phase of HIV infection, can distinguish recent (i.e., seroconversion within the past 6–12 months) from non-recent infection and have been used to evaluate the impact of large-scale HIV prevention interventions and to estimate population-level HIV incidence for the past two decades [11–13]. TRIs are designed to be performed in laboratories and results are often used in a recent infection testing algorithm (RITA) that includes supplemental information on viral load (VL), ART use, and other clinical information such as CD4 cell count, and presence of AIDS-defining illness to confirm recency status [14,15]. It is well known that recent infection assays occasionally misclassify individuals with long-term infections who are on ART or elite controller with suppressed VL (<1000 copies/mL). This misclassification occurs because antibody decays when viral load is suppressed. Therefore including viral load criteria to the RITA improves accuracy of recent infection detection [16]. RITA has been integrated into routine HIV testing services (HTS) and as part of HIV surveillance in England and other European countries [17]. In one study surveying 42 HIV specialists from 32 HIV centers on the role of RITA in clinical practice and contact tracing, 83% reported that RITA was a standard of care at their clinics and 90% endorsed the usefulness of recency results to assist with contact tracing [18]. The limiting antigen avidity enzyme immunoassay (LAg EIA) is the most commonly used TRI globally because it provides the most accurate classification of recency status compared with other TRIs [15,19], and has been used in a RITA to

assess the impact of community-based HIV prevention interventions on population-level HIV incidence and to monitor trends in national HIV incidence, which have provided vital information to inform HIV program planning [20]. In addition, a LAg EIA-based RITA has been piloted in facilities providing HIV-related services to assess the feasibility and value of different recent HIV testing approaches in routine HIV programs [20].

Despite the established role that laboratory-based TRIs and RITAs have played in informing prevention efforts and monitoring public health impact through integration in surveillance and their potential role in direct HIV service delivery, they are not easy to incorporate into routine operations due to the intensive resources required to conduct the test and long turn-around time for returning results to the referring testing facility or site. Rapid tests for recent infection (RTRIs) are novel lateral flow devices that work on the same principle as rapid HIV tests and the LAg EIA and can simultaneously diagnose HIV and differentiate between recent and non-recent HIV infection [21]. The performance of one RTRI, Asanté™ HIV-1 Rapid Recency™ assay (Sedia Biosciences, Portland, OR, USA) (hereafter referred to as the Asanté assay), has been evaluated in a controlled laboratory setting in the United States using well-characterized panels of cross-sectional specimens with known HIV serological status and recent infection status based on the LAg EIA [22,23]. The Asanté assay, whose results are available in less than 30 minutes, could provide further public health benefit when incorporated into routine HTS with both a programmatic and surveillance use case, for real-time identification of recent infection to facilitate rapid public health investigation and response [24]. The Central America Region was the first region globally, and Malawi was the first country in Africa, to pilot the use of the Asanté assay in routine clinical settings for real-time identification of transmission hot spots and targeting responses to break the chain of transmission [25].

Malawi, a county in southern Africa with high HIV prevalence, has made significant progress towards achieving the UNAIDS 90-90-90 targets. The 2015–2016 Malawi Population-based HIV Impact Assessment (MPHIA) showed that 72.7% of adult PLHIV aged 15–64 were aware of their diagnosis, 89.6% of those with known HIV diagnosis were on ART, and 91.2% of those on ART were virally suppressed [26]. The annual incidence of HIV infection among adults was 0.37 per 100 person-years in 2015 and 2016, which corresponded to approximately 30,000 new HIV infections among adults per year in Malawi [26]. MPHIA results highlighted the continuing challenges in HIV diagnosis and ongoing HIV transmissions, and the critical need for innovative approaches to close the gap in HIV diagnosis and reduce transmission.

Successful implementation of RTRI in routine HTS will require strong laboratory systems to support quality control and training. However, there are limited data on the performance of RTRI when conducted by laboratory technicians in sub-Saharan Africa where HIV burden is high and laboratory infrastructure is weak. We sought to evaluate the performance of the Asanté assay in two Malawian central laboratories among adolescent girls and young women (AGYW) seeking public-sector antenatal care (ANC) services in Malawi. The study focused on this population because they were likely to be at high risk for HIV incidence compared to the general population and would reliably seek care in the peripartum period. This study was the first step to better understand performance of the test in the field and assess the training needs as countries plan to roll out recency test at the testing sites for real-time surveillance of new infections.

## Methods

### Study design

The Malawi Recency Study was a cross-sectional study that was implemented in public health facilities in four high-HIV burden districts in Malawi, namely, Lilongwe, Blantyre, Machinga,

and Zomba (HIV prevalence ranging from 11.5% to 17.7%) from November 2017 through June 2018 [26]. The overall aim of the study was to establish a surveillance system to detect and characterize recent HIV infections among AGYW receiving ANC in the four districts. Pregnant AGYW aged 15 to 24 years who were newly diagnosed with HIV at their first ANC visit for that pregnancy were consecutively enrolled into the survey after providing informed consent. All informed consent was conducted in either the local language (Chichewa) or English. AGYW were diagnosed with HIV by the Alere Determine™ HIV-1/2 (Abbott, Abbott Park, IL, USA) and Uni-Gold™ HIV (Trinity Biotech, Wicklow, Ireland) rapid tests using Malawi's routine national HIV testing algorithm [27]. Venous blood specimens were collected from consenting HIV-positive AGYW and transported daily to a satellite laboratory in each district, namely Machinga District Hospital, Queen Elizabeth Central Hospital, Kamuzu Central Hospital, and Zomba Central Hospital, for centrifugation into plasma. Plasma samples were aliquoted and stored in -20 degrees Celsius until there were transported once weekly to two central laboratories for testing for recent infection and VL [28].

## Target sample size

The number of eligible AGYW expected to participate in the overall study in the four districts was based on routine ANC and prevention of mother-to-child transmission of HIV (PMTCT) program data collected from April to June 2016 (3 months) and adjusted to the six-month study period. We expected a similar scale of HIV positivity among AGYW, with about 400 HIV-positives, during implementation of the Malawi Recency Study.

## Laboratory methods

Testing for recent HIV infection was conducted in the Kamuzu Central Hospital and the Zomba Central Hospital laboratories located in Lilongwe and Zomba districts, respectively. Master trainers from the International Laboratory Branch, Division of Global HIV and TB, Center for Global Health, of the United States Centers for Disease Control and Prevention (CDC) trained laboratory technicians on use of the RTRI and LAg EIA. Plasma was prepared from all viable venous specimens and tested with the LAg EIA (HIV-1 LAg-Avidity EIA, SEDIA Biosciences Corporation, Portland, OR, USA), using previously described methods [29]. Specimens with normalized optical density (ODn) value ≤2.0 were classified as testing recent on the LAg EIA, otherwise they were classified as testing non-recent. The LAg EIA ODn cutoff of 2.0 corresponded to a mean duration of recent infection of 177–183 days [23,30]. The LAg-Avidity EIA runs include multiple controls and calibrator on each plate to validate the assay. In addition, all the laboratory data was reviewed by International Laboratory Branch, CDC-Atlanta.

The same plasma specimens that were tested with the LAg EIA were tested with the Asanté assay, which is based on the same principle as the LAg EIA [21]. QA measures included hands-on training of all testers and periodic testing of QC specimens. The Asanté assay is a lateral flow device with three lines, a control (C) line, a diagnostic verification (V) line and a long-term (LT) line, to distinguish recent from long-term infection [22] and can be read qualitatively by visual inspection of the test strip or quantitatively using a reader and measured in line intensity units [23]. The reader is a portable, hand-held, battery-powered device that can provide quantitative results in 20 minutes [31]. Presence of only the C line indicated the client was Asanté-assay seronegative, while presence of C and V lines indicated probable recent infection (mean duration~ 6 months). The presence of all three lines indicated diagnosis of long-term HIV infection (≥6 months). On quantitative read, V line intensity units ≥2.8 indicated that specimen was Asanté assay HIV-positive; otherwise they were Asanté assay HIV-

negative. If the LT line had <3.0 line intensity units, then the specimen was a probable recent infection by the Asanté assay; otherwise they were Asanté assay long-term. If the C line was absent on visual interpretation of the Asanté assay or the reader had line intensity units <2.8 for C line, the test was considered invalid, and the specimen was excluded [23]. For the purposes of this analysis, specimens that were recent by either assay were classified as probable recent infections. HIV VL was measured for all specimens with probable recent infection by quantifying HIV-1 RNA using the, Roche Cobas Ampliprep/Cobas Taqman (Roche Diagnostics USA, Indianapolis, IN, USA).

## Case definition for a recent and long-term HIV infection

We defined a case of recent HIV infection as a person who did not self-report current or previous ART, from whom a specimen was classified as a probable recent infection, and who was found to not be virally suppressed (defined as HIV-1 RNA ≥1000 copies/ml). A long-term HIV infection was defined as a person from whom specimen collected 1) tested long-term on either assay or 2) classified as a probable recent infection but was subsequently found to be virally suppressed (HIV-1 RNA <1000 copies/ml). VL is included in these case definitions to improve the accuracy of recent infection detection. A case with recent HIV infection implied that the infection was acquired within approximately 12 months of blood specimen collection, whereas a case with long-term HIV infection implied the infection was acquired 12 months or longer prior to blood specimen collection.

## Ethical review

The study protocol and all data collection tools were approved by the National Health Sciences Research Committee of Malawi, and the Institutional Review Boards at the CDC and the Committee for Human Research at the University of California, San Francisco.

## Data analysis

First, we calculated the sensitivity of the Asanté assay, by both visual inspection and reader, in verifying HIV seropositivity among specimens that had been determined to be HIV-positive using the Malawi rapid HIV testing algorithm. Second, we evaluated the performance of the Asanté assay reader for recent HIV infection detection by comparing it with the LAg EIA. We determined the overall correlation between the LAg EIA ODn and the Asanté assay reader results by calculating the Spearman's rho overall and stratified by laboratory (hereafter identified as laboratory A and laboratory B). Specimens collected and initially tested in laboratory A were retested in laboratory B; however, two participants' specimens did not have sufficient volume remaining and were therefore excluded from retesting. Third, we assessed the agreement between RITA using LAg EIA and RITA using Asanté assay (visual inspection and reader) in confirming recent HIV infections. We categorized specimens into confirmed recent and long-term HIV infection based on the RITA and calculated the agreement between the LAg EIA-based RITA and Asanté-based RITA. We computed the percent agreement and kappa statistics, and their associated 95% confidence intervals (95% CI). Participants without sufficient amount of specimen to complete the entire RITA were excluded from the final analysis. All statistical analyses were done using Stata 14.1 (Stata Corporation, College Station, TX, USA). Because there is no gold standard assay for recent infection, we did not calculate sensitivity, specificity, and positive and negative predictive values.

## Results

### Evaluation of RTRI HIV diagnosis verification among confirmed HIV-seropositive specimens using the national HIV testing algorithm

Overall, 380 plasma specimens from persons diagnosed with HIV using the routine Malawi testing algorithm were tested with Asanté assay. Of those 380 specimens, 377 were HIV positive (had both the C and V lines present) on visual interpretation and 376 specimens were HIV positive using the reader (had a V line intensity units of ≥2.8). Compared to the national HIV testing algorithm, the Asanté assay has a sensitivity of 99.2% (377/380) using visual interpretation and 98.9% (376/380) using the reader.

### Performance of RTRI versus LAg EIA for recent HIV infection detection

Specimens from 376 participants were tested with both the LAg EIA and the Asanté assay reader originally in two separate laboratories. The overall Spearman's rho between the two assays was 0.657 (N = 376), suggesting a moderate correlation (Fig 1). In a subset analysis stratified by laboratories A and B, the Spearman's rho between the LAg EIA ODn and the Asanté assay long-term line intensity units for specimens tested in laboratory A was 0.560 (n = 133), and the Spearman's rho for specimens tested in laboratory B was 0.747 (n = 243) (Fig 2), suggesting moderate and strong correlations, respectively.

Specimens from 131 participants originally tested for recent HIV infection by both the LAg EIA and the Asanté assay in laboratory A were retested in laboratory B. Fig 3 shows the scatterplot comparing LAg EIA ODn values and the LT line intensity units of the Asanté assay reader for overall 374 specimens (includes 131 retested samples) tested in laboratory B. The Spearman's rho between the two variables was 0.715, suggesting a strong correlation between the two assays [32,33]. The Spearman's rho between the two variables for the retested samples alone improved to 0.69 (n = 131), suggesting a moderate correlation [32,33].

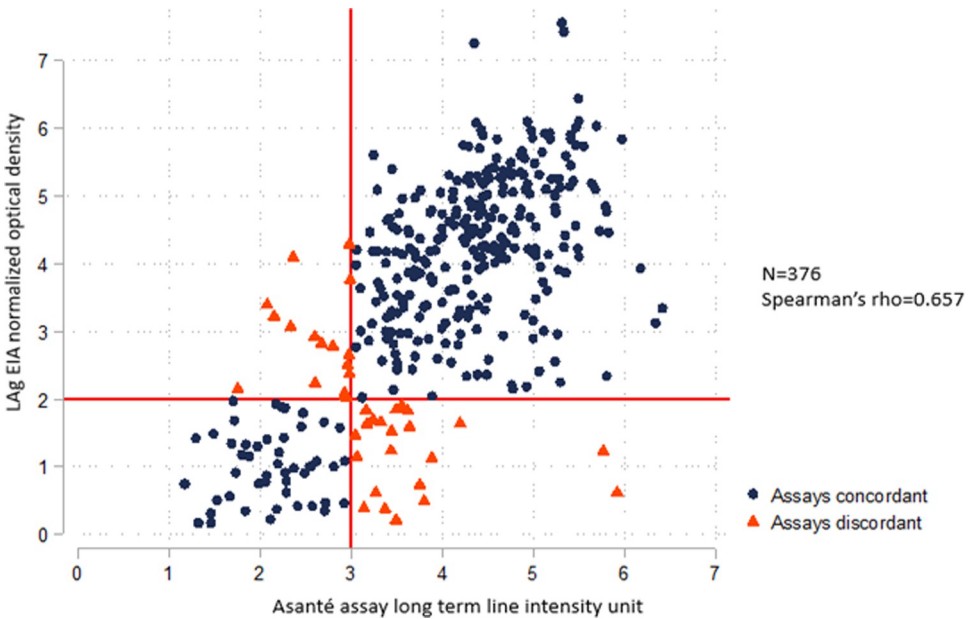

**Fig 1. Correlation between the recency component of Asanté (reader) and LAg EIA (using original data from the testing in the two separate laboratories).**

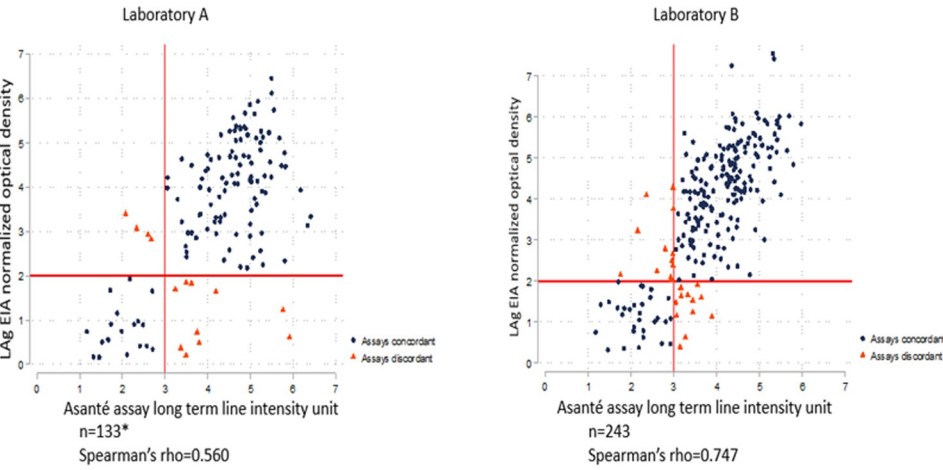

**Fig 2. Correlation between the recency component of Asanté (reader) and LAg EIA (original data stratified by laboratory).**

## Performance of RITA using Asanté assay versus RITA using LAg EIA

All enrolled participants self-reported new HIV diagnosis and presumably had no prior ART use. As outlined in Fig 4, LAg EIA was applied to 380 specimens, of which 75 (19.7%) tested recent and were classified as probable recent infection. Eighteen (24.0%) of these specimens had suppressed VL and were reclassified as long-term infections; 4 specimens that tested recent on LAg EIA did not have VL test completed. Using the Asanté assay on 376 specimens, 66 (17.6%) tested recent on the Asanté based on quantitative results from the reader and were classified as probable recent infection. Of these 18 (27.3%) had suppressed VL and were

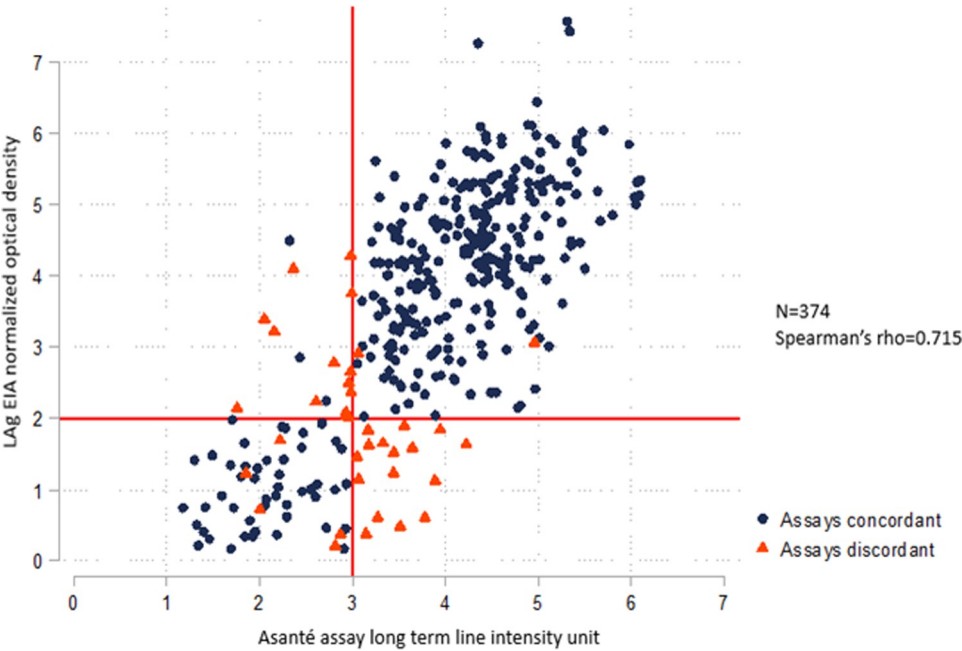

**Fig 3. Correlation between the recency component of Asanté (reader) and LAg EIA (combined data including samples retested in laboratory B).**

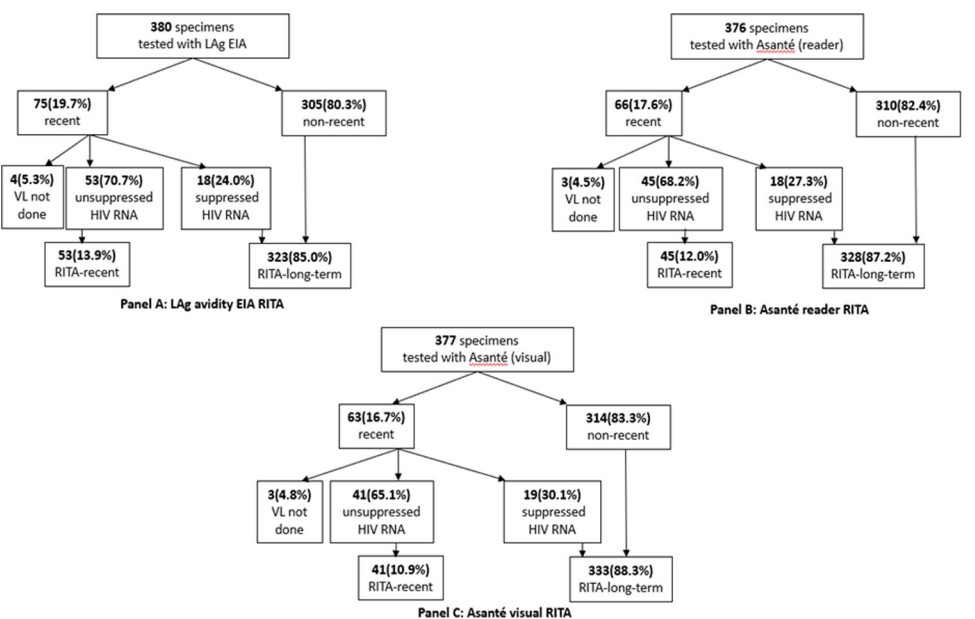

**Fig 4. RITA using LAg EIA versus RITA using Asanté assay.**

reclassified as long-term infections; 3 specimens that tested probable recent did not have VL test completed. Based on visual interpretation of the Asanté assay on 377 specimens, 63 (16.7%) tested recent on the Asanté and classified as probable recent infection; of these 19 (30.1%) had suppressed VL and were reclassified as long-term infections, 3 specimens that tested probable recent did not have VL test completed.

There were 372 specimens that were used to evaluate RITA performance based on quantitative results of the Asanté assay reader and the LAg EIA (Table 1). Of the 45 specimens that classified as confirmed recent infection on the Asanté-based RITA (based on quantitative results from the reader), 37 (82.2%) specimens were similarly classified as confirmed recent infection using LAg EIA-based RITA. Three hundred eleven (95.4%) of the 327 specimens classified as long-term infections on the Asanté-based RITA (based on the reader's quantitative results) were similarly classified as long-term infections on the LAg EIA-based RITA. The percent agreement was 93.6% (95% CI 90.6%-95.9%) with a Kappa statistic of 0.718 (95% CI 0.226–0.807) indicating substantial agreement between the Asanté-based RITA based on quantitative results from the reader and LAg EIA-based RITA [34].

Three hundred seventy-three specimens were used to evaluate RITA performance using the Asanté visual assessment and the LAg EIA (Table 2). Of the 41 specimens that had confirmed recent infection on the Asanté-based RITA (based on visual assessment), 33 (80.5%) had

**Table 1. Concordance of recent infection testing algorithm results using the LAg EIA and Asanté reader.**

| RITA using LAg EIA | Recent Infection Testing Algorithm (RITA)using the Asanté reader | | |
|---|---|---|---|
| | Recent | Long-term | Total |
| Recent | 37 | 16 | 53 |
| Long-term | 8 | 311 | 319 |
| Total | 45 | 327 | 372 |
| **Percent agreementKappa** | 93.6% (95% CI 90.6%-95.9%)<br>0.718 (95% CI 0.226–0.807) | | |

**Table 2. Concordance of recent infection testing algorithm results using the LAg EIA and Asanté visual inspection.**

| RITA using LAg EIA | Recent Infection Testing Algorithm (RITA) using Asanté visual inspection | | |
|---|---|---|---|
| | Recent | Long-term | Total |
| Recent | 33 | 20 | 53 |
| Long-term | 8 | 312 | 320 |
| Total | 41 | 332 | 373 |
| Percent agreement | 92.6% (95% CI 89.4%-95.0%) | | |
| Kappa | 0.667 (0.160–0.763) | | |

confirmed recent infection on the LAg EIA-based RITA. Overall, 312 (94.0%) of the 332 specimens that had confirmed long-term infection on the Asanté-based RITA (based on visual assessment) had confirmed long-term infection on the LAg EIA-based RITA. This comparison yielded a percent agreement of 92.6% (95% CI 89.4%-95.0%) and a Kappa statistic of 0.667 (0.16–0.763), indicating substantial agreement between the visually assessed Asanté assay RITA and Lag EIA RITA [34].

## Discussion

Our study is the first field laboratory evaluation of a RTRI in sub-Saharan Africa and showed the feasibility of using an RTRI in the laboratory among specimens collected from routine clinical services to detect recent HIV infection among AGYW newly diagnosed with HIV. The Asanté assay had high test sensitivity for detecting HIV antibodies (positive V line) among participants with confirmed HIV infection using the Malawi rapid HIV testing algorithm when conducted in a well-controlled laboratory setting. The Asanté assay achieved a test sensitivity similar to that observed by Parekh et al. in an evaluation of the Asanté assay compared against EIA plus Western blot testing on a large, well-characterized, world-wide specimen panel, conducted in a United States-based laboratory [23]. The Asanté assay also meets the sensitivity threshold for World Health Organization (WHO) prequalified HIV serology diagnostics [23,35–37]. Using the Asanté assay reader, performance for positive verification was slightly lower because one specimen had a V line intensity unit of 2.64 and fell just below reader threshold for positive verification, though the specimen was verified as HIV-positive on visual interpretation of the test strip. Comparatively, the Determine[TM] HIV1/2 Ag/Ab Combo Rapid Test had been shown in prior studies to have a sensitivity of 94% among HIV-positive pregnant women [36] and 99% among HIV-1 United States seroconverters and HIV-2-infected individuals from Côte d'Ivoire [37]. Collectively, these results highlight the potential for the inclusion of the Asanté assay as part of the national HIV testing algorithms.

The Asanté assay was also highly correlated with the LAg EIA in the detection of recent HIV infection among clinical specimens from pregnant AGYW when conducted in a central laboratory in Malawi. Initially, the 376 specimens were tested in two separate central laboratories where the specimens were received. The overall Spearman's rho showed a lower correlation than was observed in the CDC in-house evaluation of the assay on a well-characterized panel of cross-sectional specimens [23]. In sub-analyses separated by laboratory, we observed a lower correlation in laboratory A than in laboratory B. These differential results between the two laboratories suggested an issue with the testing process as opposed to the test performance of the Asanté assay. To confirm this hypothesis, we retested the samples originally received in laboratory A in laboratory B and observed improved correlation between the two assays.

These results suggest challenges in quality assurance of testing quality in laboratory A and highlight the importance of high-quality training and re-training of laboratory technicians on how to conduct the Asanté assay and laboratory quality control (QC) methods, as well as systematic and frequent laboratory data monitoring as part of routine monitoring and evaluation programs.

When the Asanté assay was incorporated into a RITA and compared against the routinely used LAg EIA RITA, there was substantial agreement between the two RITAs, whether by reader or visual assessment of the Asanté assay results. Since there is often a turnaround time of a few weeks in receiving VL test results to confirm recency status, the Asanté assay results alone may be useful to rapidly inform follow up of probable recent infection and prompt investigation into suspected transmission networks and interventions to rapidly treat and prevent further transmission to contacts.

The results of this evaluation directly led to the establishment of the United States President's Emergency Plan for AIDS Relief (PEPFAR)'s surveillance and public health response initiative through "Tracking with recency assays to control the epidemic" (TRACE), which is being implemented in over 20 countries worldwide with plans of expanding to additional countries [24]. Some countries have implemented the TRACE initiative at HTS sites with testing conducted by either HTS counselors or trained phlebotomists, while other countries have implemented the laboratory-based model where RTRI is performed at a laboratory hub. Testing at HTS sites by counselors or phlebotomists allows for rapid turn-around of results for action; however, this approach makes incorporating other tests that have longer turnaround times, such as VL, to confirm recent infection a challenge, and efforts should be made to include other routinely-collected information in patient charts, such as previous HIV diagnosis or ART use, prior to acting on probable recent infection results. In addition, staff at the HTS sites would require improved, more frequent training and site-level monitoring to succeed at conducting RTRI in even less controlled environments as shown by results of this pilot [38]. Testing at a laboratory hub has the advantage of high-quality testing but would require strong sample referral networks (given the Asante assay must be done using whole blood or plasma), high-quality training and QC methods to ensure accurate results. Moreover, results would not be readily available for immediate programmatic intervention. Given the importance of high testing quality and QC methods on RTRI performance, laboratory hub testing model where testing is conducted by trained laboratory staff could be considered in settings with large numbers of HTS sites where ensuring high-quality testing at all sites might be challenging. Early data from TRACE scale-up in various countries in sub-Saharan Africa should provide evidence on best testing models for ensuring high-quality data from routine recency testing.

Our results are comparable to the results from the evaluation of the assay on a well-characterized specimen panel from around the world [23]. A limitation is that the HIV verification component of the Asanté assay was conducted in a study sample that was comprised of individuals who were infected with HIV; as such, we were unable to evaluate the specificity of the assay in identifying HIV-uninfected individuals. Nevertheless, the high sensitivity observed was similar to assays used in current rapid HIV testing algorithms in PEPFAR-supported countries and meets the WHO prequalification threshold for HIV diagnostic tests [35–37]. Thus, the assay has the potential to provide both HIV diagnosis and determination of the recency of HIV infection, if incorporated into national testing algorithms.

Our study has demonstrated the feasibility of integrating RTRI into routine HIV services using a central laboratory testing model and established a framework for establishing recent HIV infection surveillance in Malawi. We have also shown the real-world performance of the Asanté assay and its utility in the rapid detection of recent HIV infections, which may help

identify areas with high HIV transmissions to inform effective prevention interventions to achieve HIV epidemic control.

## Acknowledgments

The authors are grateful to the Machinga District Hospital Laboratory Manager Dorothy Moyo and Queen Elizabeth Central Hospital Laboratory Manager Joseph Bitilinyu-Bangoh for their support in processing participants' specimens for recency testing, and to the Malawi Recency Study participants and staff.

## Author Contributions

**Conceptualization:** Elfriede A. Agyemang, Andrea A. Kim, Andrew F. Auld, George W. Rutherford, Bharat S. Parekh.

**Formal analysis:** Kathryn Curran, Susie Welty.

**Funding acquisition:** Andrea A. Kim, Evelyn Kim, Andrew F. Auld, Bharat S. Parekh.

**Investigation:** Elfriede A. Agyemang, Danielle Payne, Andrew D. Maher, Kathryn Curran, Evelyn Kim, Andrew F. Auld, George W. Rutherford, Bharat S. Parekh.

**Methodology:** Elfriede A. Agyemang, Andrea A. Kim, Trudy Dobbs, Innocent Zungu, Evelyn Kim, Bharat S. Parekh.

**Project administration:** Elfriede A. Agyemang, Innocent Zungu, Danielle Payne, Andrew D. Maher, Kathryn Curran, Hastings Kwalira, Henry Limula, Susie Welty.

**Resources:** Elfriede A. Agyemang, Andrea A. Kim, Danielle Payne, Andrew D. Maher, Kathryn Curran, Evelyn Kim, Hastings Kwalira, Henry Limula, Amitabh Adhikari, James Kandulu, Rose Nyirenda, Bharat S. Parekh.

**Software:** Amitabh Adhikari.

**Supervision:** Elfriede A. Agyemang, Andrea A. Kim, Trudy Dobbs, Innocent Zungu, Danielle Payne, Andrew D. Maher, Kathryn Curran, Evelyn Kim, Hastings Kwalira, Henry Limula, Bharat S. Parekh.

**Validation:** Elfriede A. Agyemang, Danielle Payne, Andrew D. Maher, Kathryn Curran, Susie Welty, Bharat S. Parekh.

**Visualization:** Elfriede A. Agyemang.

**Writing – original draft:** Elfriede A. Agyemang.

**Writing – review & editing:** Elfriede A. Agyemang, Andrea A. Kim, Trudy Dobbs, Danielle Payne, Andrew D. Maher, Kathryn Curran, Evelyn Kim, Hastings Kwalira, Henry Limula, Amitabh Adhikari, Susie Welty, James Kandulu, Rose Nyirenda, Andrew F. Auld, George W. Rutherford, Bharat S. Parekh.

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
