## [Decision Letter · Decision Letter 0]

8 Jul 2021

PONE-D-20-31833

Performance of a novel rapid test for recent HIV infection among newly-diagnosed pregnant adolescent girls and young women in four high-HIV-prevalence districts — Malawi, 2017–2018

PLOS ONE

Dear Dr. Agyemang,

Thank you for submitting your manuscript to PLOS ONE. After careful consideration, we feel that it has merit but does not fully meet PLOS ONE’s publication criteria as it currently stands. Therefore, we invite you to submit a revised version of the manuscript that addresses the points raised during the review process.

We look forward to receiving your revised manuscript.

Kind regards,

John S Lambert

Academic Editor

PLOS ONE

Journal Requirements:

"I have read the journal's policy and the authors of this manuscript have the following competing interests: As an inventor of LAg EIA and rapid test for recent infection, BSP receives a portion of royalties from the sale of these tests as per policy of the United States government. "

Reviewers' comments:

Reviewer's Responses to Questions

**Comments to the Author**

1. Is the manuscript technically sound, and do the data support the conclusions?

Reviewer #1: Yes

Reviewer #2: Yes

2. Has the statistical analysis been performed appropriately and rigorously? 

Reviewer #1: Yes

Reviewer #2: N/A

3. Have the authors made all data underlying the findings in their manuscript fully available?

Reviewer #1: Yes

Reviewer #2: Yes

4. Is the manuscript presented in an intelligible fashion and written in standard English?

Reviewer #1: Yes

Reviewer #2: Yes

5. Review Comments to the Author

Reviewer #1: Performance of a novel rapid test for recent HIV infection among newly-diagnosed pregnant adolescent girls and young women in four high-HIV-prevalence districts — Malawi, 2017–2018.

In this study, Elfriede et al. conducted a field-based laboratory evaluation of

a test for recent HIV infection (RTRI) among pregnant adolescent girls and young women (AGYW). This study evaluates the performance of this test in the field and establish a surveillance system to detect and characterize recent HIV infections among AGYW. The study is of great importance as it helps to estimate HIV incidence and monitor transmission.

The paper is well written and easy to follow with very few typos.

Nevertheless, we have 4 minor comments:

1. Usually test performance includes sensitivity and specificity of the new testing tool as compared to a gold standard: Authors only evaluated the specificity but not the sensitivity, should they not only mention specificity and not performance in the manuscript?

2. In the methods section, authors should indicate the statistical formula used to estimate the sample size in this study.

3. It is not clear to the readers why participant tested with RTRI who had LT line intensity units <3.0 and suppressed HIV-1 RNA viral load (<1000 copies/ml) were classified as non-recent infection while those who did not virally suppressed the virus (HIV-1 RNA viral load ≥1000 copies/ml) were classified as recent infection.

4. It would be desirable that the authors also make an evaluation on the estimated costs of this new test for recent HIV infection, the performance must be accompanied by the accessibility for a wide use. Low cost tests are the most needed for resource limited settings.

Reviewer #2: For discordant results laboratory A -B the impementation of these test could be compormised by the non reproducibility of the results test , it would be mandatory to have strict quality control in the labs.

6. PLOS authors have the option to publish the peer review history of their article (what does this mean?). If published, this will include your full peer review and any attached files.

Reviewer #1: No

Reviewer #2: No

---

## [Author Response · Author response to Decision Letter 0]

6 Aug 2021

We would like to thank the reviewers and editors for their review of our manuscript, “Performance of a novel rapid test for recent HIV infection among newly-diagnosed pregnant adolescent girls and young women in four high-HIV-prevalence districts — Malawi, 2017–2018”, for consideration as an original contribution to PLOS ONE.

In this revised manuscript, we have incorporated feedback received from the reviewers as appropriate. We have also included below responses to each point raised by journal editors and our reviewers. 

Journal Requirements

Comment 1. Please ensure that your manuscript meets PLOS ONE's style requirements, including those for file naming. 

Response: We have reviewed the manuscript and reformatted to meet PLOS ONE’s style requirements

Comment 2. We note that the grant information you provided in the ‘Funding Information’ and ‘Financial Disclosure’ sections do not match. When you resubmit, please ensure that you provide the correct grant numbers for the awards you received for your study in the ‘Funding Information’ section.

Response: We have ensured the grant information in the “Funding Information” and “Financial Disclosure” sections align. “This study was supported by the President’s Emergency Plan for AIDS Relief (PEPFAR) through the Centers for Disease Control and Prevention (CDC) under the terms of cooperative agreement U2GGH000977”.

Comment 3. Thank you for stating the following in the Competing Interests section:

"I have read the journal's policy and the authors of this manuscript have the following competing interests: As an inventor of LAg EIA and rapid test for recent infection, BSP receives a portion of royalties from the sale of these tests as per policy of the United States government”. 

Response: We have included that “This does not alter our adherence to PLOS ONE policies on sharing data and materials”.

Competing Interests: As an inventor of LAg EIA and rapid test for recent infection, BSP receives a portion of royalties from the sale of these tests as per policy of the United States government. This does not alter our adherence to PLOS ONE policies on sharing data and materials. All study data remain the property of Malawi Ministry of Health and can be requested by contacting Danielle Payne (ywb9@cdc.gov).

Reviewer #1

Comment 1. Usually test performance includes sensitivity and specificity of the new testing tool as compared to a gold standard: Authors only evaluated the specificity but not the sensitivity, should they not only mention specificity and not performance in the manuscript? 

Response: Recent or long-term (LT) classification with this assay is based on maturation of antibody avidity which varies from person to person. Therefore, sensitivity is not a common parameter used in the performance analysis of this assay, but specificity remains important. Here, we compared the rapid test for recent infection (RTRI) with another widely used validated laboratory-based LAg-Avidity EIA in this field evaluation. 

Comment 2. In the methods section, authors should indicate the statistical formula used to estimate the sample size in this study.

Response: The number of eligible AGYW expected to participate in the overall study in the four districts was based on routine ANC and prevention of mother-to-child transmission of HIV (PMTCT) program data collected from April to June 2016 (3 months) and adjusted to the six-month study period. We assumed a similar scale of HIV positivity, with about 400 HIV-positive cases, among AGYW during implementation of the Malawi Recency Study. We have included this in the methods section of the manuscript.

Comment 3. It is not clear to the readers why participant tested with RTRI who had LT line intensity units <3.0 and suppressed HIV-1 RNA viral load (<1000 copies/ml) were classified as non-recent infection while those who did not virally suppressed the virus (HIV-1 RNA viral load ≥1000 copies/ml) were classified as recent infection. 

Response: It is well known that recent infection assays occasionally misclassify individuals with long-term infections who are on ART or elite controller with suppressed VL (<1000 copies/mL). This occurs because antibody decays when viral load is suppressed. Adding the viral load criteria to the recency results improves accuracy of recent infection detection(1). We have included this explanation and reference citation in the introduction where we describe recent infection testing algorithm, and briefly in methods where we provide case definitions for recent and LT HIV infections. 

Comment 4. It would be desirable that the authors also make an evaluation on the estimated costs of this new test for recent HIV infection, the performance must be accompanied by the accessibility for a wide use. Low cost tests are the most needed for resource limited settings. 

Response: The technology is transferred to a commercial partner and the assay is available as a kit. Malawi was one of the first countries to pilot the assay but now it is implemented in more than 20 countries. Currently the cost of the test is about $5 per test but is likely to go down over time and more countries come on board to implement recent infection surveillance. 

Reviewer #2

Comment 1: For discordant results laboratory A -B the implementation of these test could be compromised by the non-reproducibility of the test results, it would be mandatory to have strict quality control in the labs. 

Response: We agree with this observation. We have now strengthened our training and monitoring program and implement multiple quality assurance elements in the rollout of this assay in several countries supported by the U.S. President’s Emergency Plan for AIDS Relief.

We believe we have addressed all outstanding comments from our editors and reviewers. Please don’t hesitate to contact us for any additional clarifications. We believe our findings are timely as several countries are implementing RTRI in their HIV programs and would appeal to the readership of the PLOS ONE Journal.

Sincerely, 

Elfriede A. Agyemang, MD, MPH

Division of Global HIV and Tuberculosis 

U.S. Centers for Disease Control and Prevention

Corresponding Author: 

Bharat S. Parekh, PhD (bsp1@cdc.gov) 

Elfriede A. Agyemang, MD, MPH (eagyemang@cdc.gov) 

Division of Global HIV and Tuberculosis 

U.S. Centers for Disease Control and Prevention

Reference:

1. Duong YT, Dobbs T, Mavengere Y, Manjengwa J, Rottinghaus E, Saito S, et al. Field Validation of Limiting-Antigen Avidity Enzyme Immunoassay to Estimate HIV-1 Incidence in Cross-Sectional Survey in Swaziland. AIDS Res Hum Retroviruses. 2019;35(10):896-905.

---

## [Editor Report · Decision Letter 1]

19 Dec 2021

Performance of a novel rapid test for recent HIV infection among newly-diagnosed pregnant adolescent girls and young women in four high-HIV-prevalence districts — Malawi, 2017–2018

PONE-D-20-31833R1

Dear Dr. Agyemang,

We’re pleased to inform you that your manuscript has been judged scientifically suitable for publication and will be formally accepted for publication once it meets all outstanding technical requirements.

Kind regards,

John S Lambert

Academic Editor

PLOS ONE
---

## [Editor Report · Acceptance letter]

4 Feb 2022

PONE-D-20-31833R1 

Performance of a novel rapid test for recent HIV infection among newly-diagnosed pregnant adolescent girls and young women in four high-HIV-prevalence districts — Malawi, 2017–2018 

Dear Dr. Agyemang:

I'm pleased to inform you that your manuscript has been deemed suitable for publication in PLOS ONE. Congratulations! Your manuscript is now with our production department. 

Kind regards, 

on behalf of

Dr. John S Lambert 

Academic Editor

PLOS ONE